# StrAE: Autoencoding for Pre-Trained Embeddings using Explicit Structure

**Mattia Opper**[a] and **Victor Prokhorov**[a] and **N. Siddharth**[a,b]

[a] University of Edinburgh, UK; [b] The Alan Turing Institute, UK

{m.opper,vprokhor,n.siddharth}@ed.ac.uk

## Abstract

This work presents StrAE: a Structured Autoencoder framework that through strict adherence to explicit structure, and use of a novel contrastive objective over tree-structured representations, enables effective learning of multi-level representations. Through comparison over different forms of structure, we verify that our results are directly attributable to the informativeness of the structure provided as input, and show that this is not the case for existing tree models. We then further extend StrAE to allow the model to define its own compositions using a simple localised-merge algorithm. This variant, called Self-StrAE, outperforms baselines that don't involve explicit hierarchical compositions, and is comparable to models given informative structure (e.g. constituency parses). Our experiments are conducted in a data-constrained ($\approx$10M tokens) setting to help tease apart the contribution of the inductive bias to effective learning. However, we find that this framework can be robust to scale, and when extended to a much larger dataset ($\approx$100M tokens), our 430 parameter model performs comparably to a 6-layer RoBERTa many orders of magnitude larger in size. Our findings support the utility of incorporating explicit composition as an inductive bias for effective representation learning.

## 1 Introduction

Human understanding of natural language is generally attributed to the understanding of composition. The theory is that smaller constituents (words, tokens) recursively combine into larger constituents (phrases, sentences) in a hierarchically structured manner, and that it is our knowledge of said structure that drives understanding (Chomsky, 1956; Crain and Nakayama, 1987; de Marneffe et al., 2006; Pallier et al., 2011). Compositionality can help drive efficient and effective learning of semantics as it presumes knowledge only of the immediate constituents of a phrase and the process of their composition. However, the models that have come to dominate NLP in recent years generally do not *explicitly* take this property into account.

Transformers (Vaswani et al., 2017) have the capacity to model hierarchical compositions through the self-attention mechanism, whereby tokens can come to represent varying degrees of the surrounding context. However, should such behaviour occur, it is incidental to the training process, as there is no strict requirement for tokens to represent higher-order objects, and tokens are never explicitly merged and compressed with one another. Whether Transformers are able to acquire knowledge of syntax, as understood from the linguistics perspective, remains unclear (Kim et al., 2020). However, there is evidence that these models do become increasingly tree-like with sufficient scale and training steps (Jawahar et al., 2019; Murty et al., 2022). This raises an interesting question. To what extent is this drive towards tree-likeness responsible for their representation-learning capability? Furthermore, evidence from Lake et al. (2017) suggests that incorporating appropriate inductive biases towards composition can help bridge the stark disparity between the quantity of data required for learning between ML models and humans, and allow computational models to generalise better. If it is indeed an inductive bias towards hierarchical composition that enables humans to acquire multi-level semantics efficiently, and NLP architectures generally aren't explicitly tasked with modelling such structure, *what happens when they are?*

To investigate this, we develop StrAE, a **Str**uctured **A**uto**E**ncoder framework. It takes as input a tree structure that represents a hierarchical composition process, and uses it to directly specify the flow of information from leaves to root (encoder) and back from root to leaves (decoder). To disentangle the effect of structure from other confounding factors, we constrain StrAE to only have access to the information immediately provided to

it by the input tree. This means that for a given node the model can only use information from the nodes immediately connected to it (constituents): a property we refer to as *faithfulness*. Following this, we investigate which training objectives are best suited to enable structured multi-level representation learning, and present a novel application of the contrastive loss to hierarchical tree-like structure.

To investigate the utility of compositional structure for representation learning, we employ a data-constrained setting ($\approx$10 million tokens), and evaluate on a series of tasks that measure semantics at both the sentence and word level. We compare the representations learned by StrAE to a series of baselines consisting of other tree-models which do not enforce *faithfulness* as rigidly as StrAE and a series of baselines that do not use explicit structure at all. We also verify that our performance is attributable to the form of composition expressed by the tree structures by comparing results across a range of different input structures.

Finally, to investigate how useful a simple bias towards hierarchical composition is, we extend StrAE to allow it to define its own compositional hierarchy. This variant, called Self-StrAE, utilises the learned representations and the encoder to define its own "merge" sequence, which is then employed as the tree structure for the decoder.

Our results indicate that knowledge of structure is indeed beneficial, and that surprisingly even a simple bias for hierarchical composition leads to promising results. In light of these findings, we then extended our experiments to the 100 million token range and analyse how well Self-StrAE performs in a significantly larger setting. Even more surprisingly, despite solely having 430 non-embedding parameters, Self-StrAE is able to achieve comparable performance to a 6 layer RoBERTa (Liu et al., 2019) model with 3.95 million parameters.

## 2 Model

We develop a framework (StrAE) that processes a given sentence to generate multi-level embeddings by faithfully conforming to the given structure. Intuitively, it involves *embedding* the tokens of a sentence onto the leaves of the structure, *composing* these embeddings while traversing up the structure to generate the full-sentence embedding, and then traversing back down the structure while *decomposing* embeddings from parent to children, to finally

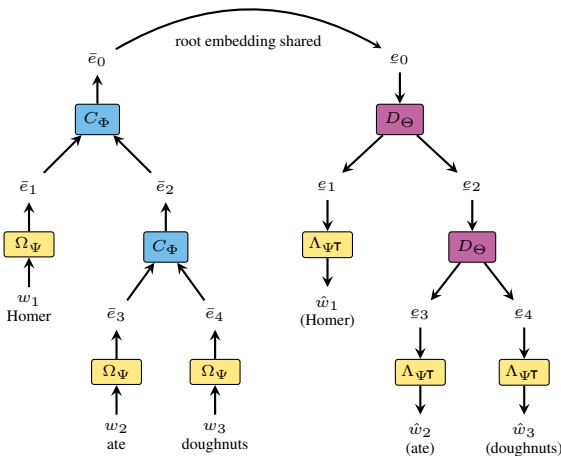

Figure 1: StrAE operation to encode and decode a sentence. Shared colours indicate shared parameters.

*recover* the sentence itself—effectively a structured autoencoder. While this generates embeddings for multiple levels of the sentence as dictated by the structure, it also generates embeddings for a node in two *directions*—composing upwards and decomposing downwards. The composition embeddings represent the local context for a node, while the decomposition embeddings represent its full context given the sequence.

Denoting embeddings $e_i \in \mathbb{R}^{N \times N}$, we distinguish embeddings formed traversing upwards and downwards as $\bar{e}_i$ and $\underline{e}_i$ respectively. Encodings for the tokens are denoted as the vertices $w_i \in \Delta^V$, in a $V$-simplex for vocabulary size $V$. Note that while the token encodings $w_i$ are effectively one-hot, reconstructions $\hat{w}_i$ can be any point on the simplex—interpreted as a distribution over vertices, and thus the vocabulary, with an argmax retrieving the appropriate token. We define the four core components of StrAE as

$$\Omega_\Psi : \Delta^V \mapsto \mathbb{R}^{N \times N} \quad \text{(embedding)}$$

$$C_\Phi : \mathbb{R}^{N \times N} \times \mathbb{R}^{N \times N} \mapsto \mathbb{R}^{N \times N} \quad \text{(composition)}$$

$$D_\Theta : \mathbb{R}^{N \times N} \mapsto \mathbb{R}^{N \times N} \times \mathbb{R}^{N \times N} \quad \text{(decomposition)}$$

$$\Lambda_{\Psi\mathsf{T}} : \mathbb{R}^{N \times N} \mapsto \Delta^V \quad \text{(indexing)}$$

where the subscripts on the functions denote learnable parameters. Table 1 shows the model components and their associated parameters. We assume square embeddings during composition and decomposition, to allow for independent channels to capture different aspects of meaning. Figure 1 shows how the model operates over a given sentence.

Table 1: Definitions of model components following § 2. Functions square and flatten transform between column vectors and square matrices. Functions hcat and hsplit perform horizontal concatenation and (middle) splitting respectively. $\sigma(\cdot)$ denotes the softmax function. $\phi$ and $\theta$ denote additive biases.

$$\Omega_\Psi(w_i) = \text{square}(w_i^\intercal \Psi) \qquad \Psi \in \mathbb{R}^{V \times N^2}$$
$$C_\Phi(\bar{e}_{c_1}, \bar{e}_{c_2}) = \text{hcat}(\bar{e}_{c_1}, \bar{e}_{c_2})\Phi + \phi \qquad \Phi \in \mathbb{R}^{2N \times N}, \phi \in \mathbb{R}^N$$
$$D_\Theta(\underline{e}_p) = \text{hsplit}(\underline{e}_p\Theta + \theta) \qquad \Theta \in \mathbb{R}^{N \times 2N}, \theta \in \mathbb{R}^{2N}$$
$$\Lambda_{\Psi\intercal}(\underline{e}_i) = \sigma(\text{flatten}(\underline{e}_i)^\intercal \Psi^\intercal) \qquad \Psi \in \mathbb{R}^{V \times N^2}$$

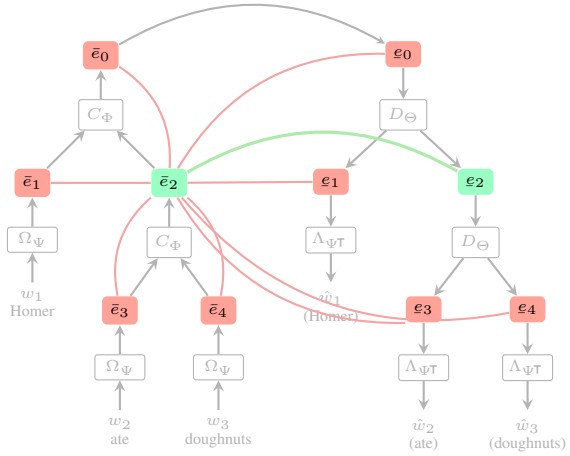

Figure 2: Contrastive objective over structure: corresponding node (green) is pulled closer, and other nodes (red) are pushed away.

## 2.1 Objectives

Given this autoencoding framework, a natural objective to employ is the cross entropy (CE), which simply measures at each leaf node how likely the target token $w_i$ is given the reconstructed distribution $\hat{w}_i$ over the vocabulary. Given sentence $s_j = \langle w_i \rangle_{i=1}^{T_j}$, this objective is formulated as

$$\mathcal{L}_{\text{CE}} = -\frac{1}{T_j} \sum_{i=1}^{T_j} w_i \cdot \log \hat{w}_i. \tag{1}$$

However, the CE objective places fairly minimal constraints on the embeddings themselves; it only requires the leaf embeddings to be 'close enough' for retrieval to be successful. The upward and downward intermediate embeddings are wholly unconstrained, and these can end up being quite different. Given we are learning embeddings for the whole tree, we would like to define an objective over all levels, not just the leaves.

For this purpose, we turn to contrastive loss (Chen et al., 2020; Radford et al., 2021; Shi et al., 2020). Contrastive loss involves optimising the embeddings of target pairs so that they are similar

to each other and dissimilar to all negative examples. To adapt its use for structured embeddings, we apply the objective so as to task the model with *maximising* the similarity between corresponding upwards and downwards embeddings $(\bar{e}_i, \underline{e}_i)$ while simultaneously *minimising* the similarity between all other embeddings. This has the additional attractive characteristic that it forces *amortisation* of the upwards embeddings $\bar{e}_i$—incorporating context from the sentences the word or phrase-segment might have occurred in, as the corresponding downward embedding $\underline{e}_i$ has full-sentence information in it. Figure 2 illustrates an example of this objective.

For a given batch of sentences $s_j$, we denote the total number of nodes (internal + leaves) in the associated structure as $M$. We construct a pairwise similarity matrix $A \in \mathbb{R}^{M \times M}$ between normalised upward embeddings $\langle \bar{e}_i \rangle_{i=1}^M$ and normalised downward embeddings $\langle \underline{e}_i \rangle_{i=1}^M$, using the cosine similarity metric (with appropriate flattening of embeddings). Denoting $A_{i\bullet}, A_{\bullet j}, A_{ij}$ the $i^{\text{th}}$ row, $j^{\text{th}}$ column, and $(i,j)^{\text{th}}$ entry of a matrix respectively, we define

$$\mathcal{L}_{\text{cont}} = \frac{-1}{2M}\left[\sum_{i=1}^M \log \sigma_\tau(A_{i\bullet}) + \sum_{j=1}^M \log \sigma_\tau(A_{\bullet j})\right] \tag{2}$$

where $\sigma_\tau(\cdot)$ is the tempered softmax (temperature $\tau$), normalising over the unspecified ($\bullet$) dimension. Note that $\mathcal{L}_{\text{cont}}$ extends to the batch setting.

Finally, we initialise our embedding matrix from a uniform distribution with the hyper-parameter $r$ denoting range—corresponding to the finding that contrastive loss seeks to promote uniformity and alignment as established by Wang and Isola (2020).

## 3 Experimental Setup

We divide our experiments into two separate sections, but both share the same overall setup for pre-training data and evaluation. For pre-training data we take a 500k sentence ($\approx$10M tokens) subset of English Wikipedia and 40k sentence development set. We restrict our pre-training data to this scale in order to measure our efficiency hypothesis presented in the introduction. For evaluation, we assess on three categories of tasks: word level semantics, sentence level semantics, and sentence pair classification, with each aiming to capture separate areas of semantic understanding. On the word level we use Simlex (Hill et al., 2015), Wordsim

Table 2: Overview of Evaluation Tasks

| Name | Level | Task | Requires Classifier? |
|------|-------|------|---------------------|
| SimLex | Word Level | Semantic Similarity | No |
| WordSim S | Word Level | Semantic Similarity | No |
| WordSim R | Word Level | Semantic Relatedness | No |
| STS 12 | Sentence Level | Semantic Similarity | No |
| STS 16 | Sentence Level | Semantic Similarity | No |
| STS B | Sentence Level | Semantic Similarity | No |
| SICK R | Sentence Level | Semantic Relatedness | No |
| MRPC | Sentence Level | Paraphrase Detection | Yes |
| RTE | Sentence Level | Textual entailment | Yes |

S and Wordsim R (Agirre et al., 2009). On the sentence level, we use three tasks from the STS suite (Agirre et al., 2016, 2012; Cer et al., 2017), the SICK relatedness dataset (Marelli et al., 2014), and for sentence pair classification tasks we use RTE and MRPC taken from the GLUE Benchmark (Wang et al., 2019a). Table 2 provides an overview. Note that a subset of the tasks distinguish between *similarity* and *relatedness*. Briefly, the former measures semantic similarity, as between "running" and "dancing", as both words act as verbs. Relatedness measures semantic relationships such as between "running" and "Nike" where the words often co-occur together, but belong to different grammatical categories. Finally, we note that Simlex only measures similarity at the *exclusion* of relatedness.

All tasks apart from the final two classification tasks are measured using the Spearman correlation of the cosine similarity of model embeddings for each pair and human judgements; classification tasks are measured using accuracy. To emphasise the impact of the pre-training with structure, we do not fine-tune any models for the evaluation tasks, instead keeping them frozen. Where a classifier is required, we fine-tune only a task-specific classification head consisting of a FFN with a 512 dimensional hidden layer and an intermediate Tanh activation function. We choose this setup to match the GLUE Benchmark. For all experiments, we pre-train the models across 5 random seeds and present the averaged performance. Downstream classifiers are themselves also trained across 5 random seeds and the average reported. Additionally, for all experiments we set the embedding dimension to 100. Finally, for all models trained on the word level we filter the vocabulary to exclude words occurring fewer than two times in the data.

## 4 Comparing Tree Models

Here, we compare StrAE with two existing tree-architectures. These are the IORNN (Drozdov et al., 2020, 2019; Le and Zuidema, 2014) and

the Tree-LSTM (Tai et al., 2015). Both architectures take structure as input and are able to traverse through the tree to learn representations. However, they differ in the constraints they impose on information flow. We conduct our experiments along two different axes: how well do the representations perform, and to what extent do the models discriminate between input structures?

To achieve these evaluations, we parse our pre-training set into three kinds of structure. The first are constituency trees extracted from CoreNLP (Manning et al., 2014) and binarised using NLTK (Bird et al., 2009). The two other kinds are purely right-branching and balanced binary trees, which we extract using standard algorithms. The resulting structures are then converted to DGL graphs (Wang et al., 2020). Hyper-parameter descriptions for each model can be found in Appendix A.

**Baselines**

The Inside-Outside Recursive Neural Network (IORNN) processes data hierarchically, working from the "inside" to the "outside". At each node in the tree, an IORNN maintains two vectors: The inside vector $\bar{e}$, which represents the local meaning of a given node, obtained by composing up the tree. The outside vector $\underline{e}$, represents the context for the given node obtained by decomposing down the tree. While superficially similar to StrAE, the models differ in an important aspect. For given parent $p$ and children $c1$, $c2$ the outside representation:

StrAE:

$$\underline{e}_{c1}, \underline{e}_{c2} = \mathrm{hsplit}(\underline{e}_p \Theta)), \text{ with } \Theta \in \mathbb{R}^{N \times 2N}$$

IORNN:

$$\underline{e}_{c1} = \tanh([\underline{e}_p; \bar{e}_{c2}]\Theta), \text{ with } \Theta \in \mathbb{R}^{2N \times N}$$
$$\underline{e}_{c2} = \tanh([\underline{e}_p; \bar{e}_{c1}]\Theta)$$

where $[\cdot; \cdot]$ denotes concatenation. In StrAE the outside representation solely depends on the parent and therefore enforces a compression bottleneck at the root of the structure. No other information may be shared from the composition process, and all $\underline{e}$ embeddings are created recursively based on the root. The outside vector for IORNN is derived from both the parent and the *inside* vector of a given child's sibling node. As the root has no siblings or parent nodes, the outside vector consists of a global bias parameter, intended to represent the context of the whole pre-training corpus. Consequently, IORNN does not enforce a compression bottleneck and information flows between both the local compositional (bottom up) and the global decompositional (top down) contexts.

Table 3: Comparison of StrAE, IORNN and Tree-LSTM embedding performance on our evaluation suite. Higher is better. All tasks that use Spearman's $\rho$ have the result * 100 and are marked with a †. Score represents the average across all tasks. All models were trained over five random seeds using constituency parses as input. The pre-training objective is indicated by C or CE, representing contrastive loss or cross entropy respectively. Only results where there is no standard deviation overlap between model performance are bolded.

| Model | Simlex † | Wordsim S † | Wordsim R † | STS 12 † | STS 16 † | STS B † | SICK R † | MRPC | RTE | Score |
|---|---|---|---|---|---|---|---|---|---|---|
| StrAE C | 15.54 ± 0.25 | 56.1 ± 0.47 | **43.77 ± 1.01** | **34.59 ± 2.58** | **50.4 ± 0.97** | **41.83 ± 2.23** | **50.3 ± 0.88** | 67.24 ± 0.39 | 56.16 ± 1.59 | **46.21** |
| StrAE CE | 17.57 ± 1.08 | 50.72 ± 2.97 | 36.73 ± 6.19 | 5.02 ± 1.1 | 25.86 ± 1.03 | 7.76 ± 0.95 | 39.06 ± 1.51 | 67.48 ± 0.3 | 53.6 ± 0.46 | 33.76 |
| IORNN C | 1.9 ± 0.25 | 27.29 ± 0.57 | 11.29 ± 0.5 | -4.2 ± 0.97 | 17.47 ± 1.14 | 1.87 ±0.55 | 30.48 ± 0.55 | 67.08 ± 0.16 | 55 ± 1.44 | 23.13 |
| IORNN CE | **24.36 ± 0.6** | 57.53 ± 1.59 | 36.55 ± 1.78 | -0.17 ± 0.92 | 22.68 ± 0.98 | 5.04 ± 0.55 | 38.41 ± 0.63 | 67.24 ± 0.63 | 54.28 ± 0.55 | 33.99 |
| Tree-LSTM C | 6.45 ± 0.4 | 37.39 ± 1.2 | 20.6 ± 1.49 | -1.34 ± 1.33 | 23.95 ± 0.45 | 4.25 ± 1.11 | 32.9 ± 0.55 | 68.16 ± 0.32 | 52.76 ± 1.09 | 27.24 |
| Tree-LSTM CE | 14.23 ± 2.01 | 48.7 ± 3.22 | 33.24 ± 3.02 | -1.66 ± 1.29 | 19.38 ± 1.21 | 6.7 ± 1.04 | 34.55 ± 1.15 | 68 ± 0.0 | 52.5 ± 1.98 | 30.59 |

Our second baseline, the Tree-LSTM, is a recursive variant of the LSTM (Hochreiter and Schmidhuber, 1997). The main difference is that inputs are processed recursively rather than recurrently. At each node, the inputs for the cell are the children's hidden and cell states. Here the flow of information differs to StrAE because while StrAE has to compress embeddings according to the order dictated by the input structure, the Tree-LSTM is able to selectively retain information from lower down in the tree according to the cell state. Tree-LSTMs can be applied both bottom up (Choi et al., 2017; Maillard et al., 2017; Tai et al., 2015) as encoders, and top down as decoders (Dong and Lapata, 2016; Xiao et al., 2022). Consequently, they can be pre-trained in the same way as StrAE. Parameter counts for all models can be found in Table 5.

**Performance on Constituency Parses**

We first evaluate model performance purely on the constituency parse structure, as in theory this should be the most informative. Results can be found in Table 3. We train all models using both objectives for the sake of parity with StrAE. Our results demonstrate that while all models are able to capture word level semantics to some degree, it is only StrAE coupled with the contrastive objective that is able to extend this to sentence level semantics, though StrAE with the cross entropy objective still performs better than the other architectures in this regard. These results indicate that enforcing *faithfulness* is beneficial in learning multilevel representations. It also follows that the contrastive objective is beneficial for StrAE as it is directly applied to all nodes, and therefore directly optimises the root's relations to other sentences. In the classification tasks, there appears to be little difference between architectures.

It is clear that the contrastive objective is only useful when the model imposes constraints on information flow, as both baselines do not benefit from it. The objective asks the model to reconstruct each node embedding such that it is distinctive from all other embeddings in the batch. If the LSTM is retaining information in its cell state, it makes this distinction difficult to enforce. IORNN faces difficulties in two regards: the outside representation of the root is a global parameter which makes distinction significantly more difficult, and the sharing of information between sibling nodes. We tested the effects of removing the sharing of information and still found little improvement, possibly because the degree of information sharing between inside and outside representations is so high that it renders the requirement for meaningful compression void.

Evidence of IORNN's reliance on the information sharing from inside sibling nodes can be seen in its Simlex performance with the cross entropy objective. Simlex actively penalises capturing semantic relatedness, which in the case of IORNN is information that would be provided by the outside vector. Its poor performance indicates that IORNN is not using the parent outside representation as significantly when reconstructing embeddings, which may explain the difficulties on the sentence level tasks as the model can simply try to predict a word given its immediate left or right neighbour.

**Performance Across Input Structures**

We also evaluate how dependent the performance of each model is on input structure. We compare the performance of each model between constituency parses, right-branching and balanced binary trees as input. Plots of these results are in Fig. 3, and the full tables can be seen in Appendix D. StrAE is dependent on input structure for performance, particularly when using the contrastive objective, though this still holds true across all task areas even for cross entropy. On the other hand, the baseline models are not, though this varies in degree. IORNN trained with cross entropy actually performs best with right-branching

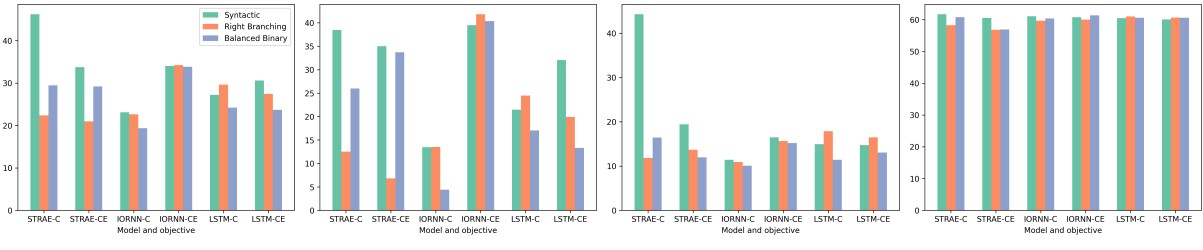

(a) Overall Score     (b) Lexical Semantics     (c) Sentence Level Semantics     (d) Classification

Figure 3: Average performance for models on different task areas by structure type, higher is better.

structure, though this is skewed by word level performance. Conversely, the LSTM appears to discriminate, but this is solely due to word level results and does not extend to other areas of evaluation.

**Summary**

StrAE is tasked with taking a sequence and compressing it into a single representation through ordered merges as dictated by the input tree. Each non-leaf node acts as a compression bottleneck. As a result, the contribution of each input token to the final root representation is directly dependent on the merge order. This makes StrAE *structure sensitive*. When coupled with contrastive loss, StrAE must learn embeddings such that similarity is dependent on how sequences compose. This is because the objective is now over all nodes, and all non-leaf node embeddings are defined by merge order. Secondly, it requires that the merge order have some degree of consistency to enable reconstruction, something which purely right-branching and balanced binary trees do not provide. It is the combination of strict compositional bottlenecking and the contrastive objective that enables StrAE to learn effective multi-level representations and serve as a probe for the utility of structure, unlike the tree structured baselines.

## 5 Comparison to Unstructured Baselines

Here, we compare StrAE to a series of baselines that are not provided any form of parse tree as input. The aim is to evaluate the usefulness of explicit structure against other inductive biases. We also introduce a variant of StrAE called Self-StrAE that is not provided structure as input but must learn its own mode of composition. It serves as a measure of the utility of an explicit compositional bottleneck.

**Self-StrAE**

Self-StrAE (Self-Structuring AutoEncoder) modifies the encoder so it uses greedy agglomerative clustering in order to decide which tokens to compose. Unlike StrAE where the order of compositions is defined by the input structure, in Self-StrAE

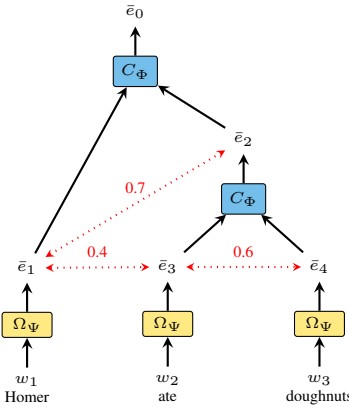

Figure 4: Self-StrAE composition. Red dotted arrows indicate cosine similarity between nodes.

this is dictated by the embeddings. Self-StrAE orders compositions according to cosine similarity between adjacent node embeddings in the frontier, merging the argmax at each step. Figure Fig. 4 shows this process. In the figure, the similarity between 'ate' and 'doughnuts' is greater than that of 'ate' to 'Homer', so the model first merges 'ate doughnuts' into a single embedding using the composition function. At the next step, the model merges 'Homer' and 'ate doughnuts' into a single embedding and arrives at the root. The algorithm is provided in Appendix B.

We save the merge history in an adjacency matrix so that by the time the encoder reaches the root node, the adjacency matrix represents the whole tree over the input sequence. We then convert the adjacency matrix into a DGL graph, and pass that as input to the decoder. The decoder operates exactly the same as in vanilla StrAE, only that in this case the input graph is defined by the encoder's merge history as opposed to, e.g., a syntactic parser.

**Baselines**

We selected Fasttext (Bojanowski et al., 2016), a Bi-LSTM (Hochreiter and Schmidhuber, 1997) and RoBERTa (Liu et al., 2019) as baselines. Fasttext leverages distributional semantics and subword information to learn word embeddings, but only op-

Table 4: Comparison to unstructured baselines. Higher is better. All tasks that use Spearman's $\rho$ have the result * 100 and are marked with a †. Score represents the average across all tasks. All models were trained over five random seeds. Only results where there is no standard deviation overlap between model performance are bolded.

| Model | Simlex † | Wordsim S † | Wordsim R † | STS 12 † | STS 16 † | STS B † | SICK R † | MRPC | RTE | Score |
|---|---|---|---|---|---|---|---|---|---|---|
| StrAE Syntactic | 15.54 ± 0.25 | 56.1 ± 0.47 | 43.77 ± 1.01 | 34.59 ± 2.58 | 50.4 ± 0.97 | 41.83 ± 2.23 | 50.3 ± 0.88 | 67.24 ± 0.39 | 56.16 ± 1.59 | **46.21** |
| Self-StrAE BPE | 13.04 ± 0.07 | 48.19 ± 1.03 | 45.47 ± 0.83 | 34.42 ± 0.73 | 49.93 ± 0.38 | 36.68 ± 0.64 | 51 ± 0.3 | 68.91 ± 0.2 | 54.42 ± 0.49 | 44.67 |
| Self-StrAE Word | 18.21 ± 0.12 | 53.52 ± 0.39 | **48.76 ± 0.86** | 33.22 ± 0.38 | 49.91 ± 0.35 | 27.97 ± 0.2 | **52 ± 0.52** | 68.13 ± 0.1 | 54.53 ± 0.53 | 45.14 |
| Fasttext | **25.8 ± 0.16** | 50.8 ± 0.24 | 29.18 ± 0.22 | 4.35 ± 0.6 | 32.43 ± 0.1 | 16.93 ± 0.09 | 41.58 ± 0.05 | 69.1 ± 0.12 | 54.4 ± 0.67 | 36.06 |
| Bi-LSTM Word | 23.74 ± 1.17 | **61.94 ± 2.44** | 41.46 ± 1.73 | 11.18 ± 1.1 | 34.86 ± 0.86 | 13.46 ± 1.11 | 42.36 ± 0.19 | 68.95 ± 0.38 | 54.09 ± 0.95 | 39.12 |
| Bi-LSTM BPE | 12.88 ± 0.77 | 39.46 ± 2.17 | 32.22 ± 3.07 | 9.22 ± 0.87 | 33.78 ± 1.66 | 14.06 ± 2.01 | 40.36 ± 0.27 | 68.21 ± 0.41 | 53.9 ± 0.96 | 33.79 |
| RoBERTa | 9.92 ± 2.7 | 26.6 ± 6.71 | 6.2 ± 3.81 | 29.48 ± 3.28 | 50.88 ± 1.11 | 38.36 ± 1.9 | 49.58 ± 0.91 | 69.19 ± 0.27 | 54.86 ± 0.49 | 37.23 |

erates over a fixed window size. The Bi-LSTM allows us to measure the utility of sequential vs hierarchical information processing. Finally, RoBERTa serves as a suitable Transformer baseline, because it only utilises the MLM objective, rather using NSP as with BERT (Devlin et al., 2019), and this has been shown to be more robust. Furthermore, it does not require additional data labelling, such as identifying which sentences follow each other, which provides greater parity with Self-StrAE and Fasttext as both models do not have such additional labels as input. For both models, we set the embedding dimensionality to 100 to match StrAE and set the number of attention heads in RoBERTa to 10 in order to match StrAE's channels. We set the number of layers in RoBERTa to 6 as, in a data constrained setting there was no additional benefit observed with a greater number, and the parameter disparity between StrAE and RoBERTa is already substantial (see Table 5 for parameter counts for both the tree and unstructured baselines). The other hyperparameter details for all baselines can be found in Appendix A. To produce sentence embeddings we take the mean of the word embeddings with Fasttext and the mean of the final layer token representations for RoBERTa. To produce word embeddings for RoBERTa we follow the lessons from Jawahar et al. (2019); Vulic et al. (2020), which state that lexical information is contained in the lower layers. For cases where a word is broken into multiple subwords, we take the average of the embeddings from layers 0-2. Where a word is present in the vocabulary, we simply use its embedding, as there is no context to provide it with. In the case of the Bi-LSTM we produce sentence embeddings by passing the concatenated final hidden states for both directions through a learned linear layer in order to produce a single 100-dimensional embedding. We found this to lead to improved performance compared to simply using the concatenation. The same strategy is used

Table 5: Number of parameters for our models and baselines. Following convention we exclude the embedding matrix (and LM head if applicable) from the count.

| Model | (Self-)StrAE | IORNN | Tree-LSTM | Bi-LSTM | RoBERTa |
|---|---|---|---|---|---|
| Params | 430 | 40,400 | 260,600 | 181,800 | 3,950,232 |

to produce word embeddings in the case where it is broken into multiple subwords. We contrast these models with StrAE trained on constituency parses with contrastive loss, Self-StrAE trained on the word level using the same vocabulary as before, and finally Self-StrAE starting from the subword level. We take the vocabulary from the same BPE (Sennrich et al., 2016) tokeniser used by RoBERTa (minus the special tokens) with a total size of 25000. We train Self-StrAE with the contrastive objective because this proved significantly more effective.

**10 Million Tokens**

As shown in Table 4, while StrAE performs best overall, Self-StrAE is able to achieve comparable performance across both the word and sentence level. Fasttext performs best on Simlex, but this is likely to be for the same reasons as IORNN i.e., capturing similarity at the expense of relatedness. This is also indicated by its comparative lower Wordsim R performance. On both other lexical semantics tasks StrAE and Self-StrAE both outperform it. Fasttext also struggles on the sentence level. Similarly, the Bi-LSTM performs well on lexical semantics, but struggles at capturing higher levels. This is evidenced both by the STS results and lower lexical performance of the BPE Bi-LSTM compared with Self-StrAE. RoBERTa performs comparably to StrAE on STS, but struggles on the word level. This might simply be because Transformers aren't designed to learn static lexical embeddings, but could also be the result of our data-constrained pretraining setting, as there was significant variability between seeds. Consequently, we conducted a final experiment using a significantly larger pre-training set to assess the impact of scale.

Table 6: WikiText-103 Results. Higher is better. All tasks that use Spearman's $\rho$ have the result * 100 and are marked with a †. Score represents the average across all tasks. All models were trained over five random seeds. Only results where there is no standard deviation overlap between model performance are bolded.

| Model | Simlex † | Wordsim S † | Wordsim R † | STS 12 † | STS 16 † | STS B † | SICK R † | MRPC | RTE | Score |
|---|---|---|---|---|---|---|---|---|---|---|
| RoBERTa | **19.28 ± 1.02** | 46.38 ± 3.18 | 26.12 ± 3.09 | 35.38 ± 1.47 | 52.64 ± 1.11 | 39.74 ± 1.04 | 50.68 ± 0.43 | **69.74 ± 0.42** | 53.71 ± 0.5 | 43.76 |
| Self-StrAE | 13.41 ± 0.66 | 47.06 ± 0.61 | **42.53 ± 1.71** | **46.64 ± 0.23** | 52.08 ± 0.37 | 40.59 ± 0.83 | **51.94 ± 0.4** | 68.35 ± 0.46 | **55.87 ± 0.3** | **46.39** |

**100 Million tokens**

For this experiment, we turned to the WikiText-103 benchmark dataset (Merity et al., 2016). WikiText-103 consists of 103 million tokens, with an average sequence length of 118. Each input sequence corresponds to an article rather than a sentence, as in our original dataset. We set the maximum sequence length to 512 and train a subword tokeniser on the training set with a maximum vocabulary of 25000. This vocabulary is used for both RoBERTa and Self-StrAE. As shown in Table 6, under this setting, RoBERTa improves significantly on the word level. RoBERTa also shows improvement on the sentence level, though these are less pronounced as the model was already performing well on these tasks. Surprisingly, Self-StrAE is able to achieve comparable (and in some cases better) performance than the RoBERTa model, despite having orders of magnitude fewer parameters. We can only attribute Self-StrAE's performance to the inductive bias behind it: that the model must perform hierarchical compositions of its input sequence. Which we believe speaks strongly to its merits.

**Summary**

Comparison with the unstructured baselines shows that *explicitly* incorporating hierarchical compositions to be beneficial for multi-level representation learning. With Self-StrAE we show that these benefits do necessitate an external parser for preprocessing, and can largely be achieved through an inductive bias for explicit merging.

## 6 Related Work

**Recursive Neural Networks:** StrAE belongs to the class of recursive neural networks (RvNNs). First popularised by Socher et al. (2011, 2013), who employed RvNNs to perform fine-grained sentiment analysis, utilising tree structure to overcome the deficits of bag of words models. These early successes inspired the creation of successor frameworks like the IORNN (Le and Zuidema, 2014) and Tree-LSTM (Tai et al., 2015) we used as baselines.

**Learning Tree Structure:** The induction of structure has long been a goal of NLP. Early pivotal work on corpus based induction was performed by Klein and Manning (2004); Petrov et al. (2006), and enabled the development of work on tree-structured models that followed. A recent prominent approach is the C-PCFG (Kim et al., 2019).

**Induction Through Representation Learning:** DIORA and subsequently S-DIORA (Drozdov et al., 2020, 2019) are models which induce structure using the IORNN. Instead of providing a fixed tree as input, they train by using dynamic programming over the set of all possible trees based on how well IORNN is able to reconstruct the input sequence. At test time, they use CKY to extract the highest scoring tree as the parse, which achieved SOTA on unsupervised consituency parsing. While these do learn representations, they solely utilise it to enable unsupervised parsing.

**Learning Task Specific Tree Structures:** Inspired by Socher et al. (2011) prior work has sought to learn trees for supervised tasks (Choi et al., 2017; Maillard et al., 2017; Yogatama et al., 2017), under the assumption that a particular task requires its own form of composition. They achieved success, but all use the Tree-LSTM, which was shown to be largely structure agnostic in the supervised setting (Shi et al., 2018); a finding we confirm in this work.

**The Utility of Structure:** Prior work has also sought to augment the Transformer architecture with tree structure. Wang et al. (2019b) modify self-attention so that it may only operate within constituents. Zanzotto et al. (2020) provide a module that represents the parse tree for an input and allows the Transformer to attend to it. Sartran et al. (2022) perform pre-training using linearised trees. In all cases, it was found that structure aided with language modelling perplexity and generalisation.

**Structure and Representation Learning:** This area, the focus of our paper, remains largely underexplored. Prior work has solely examined the word level, using dependency parses to define the context neighbourhood for a given word. The first work to do this was Levy and Goldberg (2014) and achieved promising results, however, they faced issues with vocabulary size becoming intractable. Vashishth et al. (2019) alleviated this issue through the use of GCNs (Kipf and Welling, 2016).

## 7 Discussion and Future Work

We establish two findings. Firstly, defining representation similarity through composition is useful, and secondly, asking a model to arrange its own composition is a powerful inductive bias. Neither of these findings are limited in their application to the architecture presented in this paper. The requirements are: an explicit merge operation and an objective that optimises representations across all levels. As long as these conditions are met, the findings are in theory applicable to any number of architectures. For future development, the natural next step is to examine what happens when we allow for a significantly more flexible models to dictate their own compositions. Transformers can be naturally adapted to incorporate such a bias, and we believe this holds considerable promise for future work, especially in light of recent findings that incorporating compression bottlenecks in Transformers is beneficial (Nawrot et al., 2023, 2021). Extensions need not be limited to the Transformer architecture, either. Recurrent and recursive neural networks share many similar features, and given the recent resurgence of RNNs (Orvieto et al., 2023; Peng et al., 2023) there may also be promise in extending research in this direction.

## Limitations

The particular Self-StrAE presented in this paper is a considerably limited model. It is minimally parameterised (see Table 5), locally greedy and makes uncontextualised decisions about which nodes to merge. The mode of composition it learns is certain to be suboptimal as a result of this, and it speaks to the strength of the inductive bias that it is able to perform at all. The structure it learns certainly doesn't resemble syntax as we understand it (example trees can be found in Appendix C), and neither did we expect it to. A significantly more flexible model would likely be required in this regard. Even then, it is possible that the form of composition a model learns with respect to its training objective may deviate substantially from our expectations of how compositional structure should look. Secondly, the aim of this paper is to conduct basic research into the benefits of composition and not to outperform the state of the art. We believe there is promise in pursuing further research that may eventually lead to improvements over SOTA, but we leave this to future work.

## Ethics Statement

The aim of this paper is to examine whether explicitly incorporating hierarchical compositions can be beneficial. The eventual intended goal is to make machine learning models less resource- and data-intensive. We believe this can provide considerable benefits in making research more accessible.

## Acknowledgements

MO was funded by a PhD studentship through Huawei-Edinburgh Research Lab Project 10410153. VP was funded by a research grant from the Edinburgh Lab for Integrated Artificial Intelligence (ELIAI).

We also wish to thank Eduardo Ponti, Seraphina Goldfard-Tarrant, Ivan Titov, Tom Sherborne, Vivek Iyer, Frank Mollica and Ivan Vegner for their valuable comments and feedback.

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

## A    Training Data and Hyper-parameters

We trained each StrAE (and the tree baselines) model for 15 epochs (sufficient for convergence) using the Adam optimizer with a learning rate of 1e-3 for cross entropy and 1e-4 for contrastive loss, using a batch size of 128. We applied dropout of 0.2 on the embeddings and 0.1 on the composition and decomposition functions. The temperature hyper-parameter for the contrastive objective was set to 0.2 and the r value 0.1. These settings were obtained by a grid search over the values r= 1.0, 0.1, 0.01, batch size = 128, 256, 512, 768, $\tau = 0.2$, 0.4, 0.6, 0.8 and learning rate 1e-3, 5e-4, and 1e-4. The Fasttext baseline was trained for 15 epochs with a learning rate of 1e-3, and a window size of 10. Self-StrAE was trained using the same hyperparameters as StrAE, except on Wikitext-103 where we lowered the learning rate to 5e-5 and increased $\tau$ to 0.6, and decreased r to 0.0001. RoBERTa was trained for 100 epochs, with a 10% of steps used for warmup, a learning rate of 5e-5 and a linear schedule. We used relative key-query positional embeddings. The Bi-LSTM was trained for 15 epochs with learning rate 1e-3, batch size 128 and dropout of 0.2 applied to the embeddings and output layer.

## B    Self-StrAE Algorithm

---

**Algorithm 1** Self-StrAE Agglomerative Compose

---

**Input:** Node frontier $\{e_n\}_{n=1}^N$, composition ($\circ$) pairwise node similarity $\text{CSIM}(e, e')$

1: $\mathcal{A} \leftarrow \{e_n\}_{n=1}^N$     ▷ initialise frontier
2: $\mathcal{T} \leftarrow \varnothing$     ▷ initialise tree merges
3: **while** $|\mathcal{A}| > 1$ **do**
4:    $e_i^\star, e_{i+1}^\star \leftarrow \underset{e_i,e_{i+1}\in\mathcal{A}}{\arg\max}\,\text{CSIM}(e_i, e_{i+1})$
        ▷ choose closest adjacent pair in $\mathcal{A}$
5:    $\mathcal{A} \leftarrow \mathcal{A} \setminus \{e_i^\star, e_{i+1}^\star\}$  ▷ remove from frontier
6:    $e_p^\star = \circ(e_i^\star, e_{i+1}^\star)$    ▷ compute composition
7:    $\mathcal{A} \leftarrow \mathcal{A} \cup_i \{e_p^\star\}$   ▷ insert into frontier at $i$
8:    $\mathcal{T} \leftarrow \mathcal{T} \cup \{(i, i+1)\}$  ▷ record merge location
9: **return** Merge order $\mathcal{T}$

---

## C    Tree Statistics and Examples

Self-StrAE learns trees which are not purely right-branching, balanced binary, or purely random. We parse our development split using Self-StrAE BPE models pre-trained on the 10M corpus. The split itself consists of 40k sentences with an average length of 23.58. The trees from Self-StrAE have an average depth of 9, compared with 23 (rounding up for simplicity) for right-branching trees, and 5 for balanced binary trees. Self-StrAE trees exhibit a slight preference for right-branching, with each non-leaf node on average having fewer left than right successors 60% of the time.

However, the best way to get a sense for the kind of structures the model learns is by looking at examples shown on the following pages. Looking at the examples in Fig. 5 and Fig. 6 the model has learned some sensible pattens. For example, it has learned to segment sentences with embedded clauses or conjunctions into their constituent parts (e.g. Fig. 5a,e,g and Fig. 6a). However, the trees frequently exhibit attachment errors that we hypothesize are the result of structure being determined by co-occurrence frequency. For example, in Fig. 6a the model merges [will be] as its own constituent rather than the correct parse of [will [be cancelled]]. Which is likely because "will" and "be" co-occur much more frequently than any instance of "be" + passive form of a given verb. Similar behaviour can be found all throughout the examples in Fig. 5 and Fig. 6. Given the simplicity of the model it is unsurprising that Self-StrAE is unable to learn deeper rules, however, it would be interesting to determine to what extent this is also a function of the data it is trained on. Recent work has shown that transcribed child directed speech leads transformer models to learn grammar more efficiently (Huebner et al., 2021; Mueller and Linzen, 2023). Transcribed CDS is far less regular than Wikipedia and *may* cause the model to avoid simpler heuristics like bigram frequency.

Finally, there are cases where the model seems to be learning totally implausible structures. We are yet to determine the root cause of this, but include examples in Fig. 7 for the sake of transparency.

## D    Performance by Structure Type

We show in Table 7 a comparison of performance for different structure types and objectives used in this work. As discussed earlier, the objectives used here are contrastive (C) and standard cross-entropy (CE), and the structure types explored are the syntactic, purely right branching (RB) and the balanced binary (BB) trees.

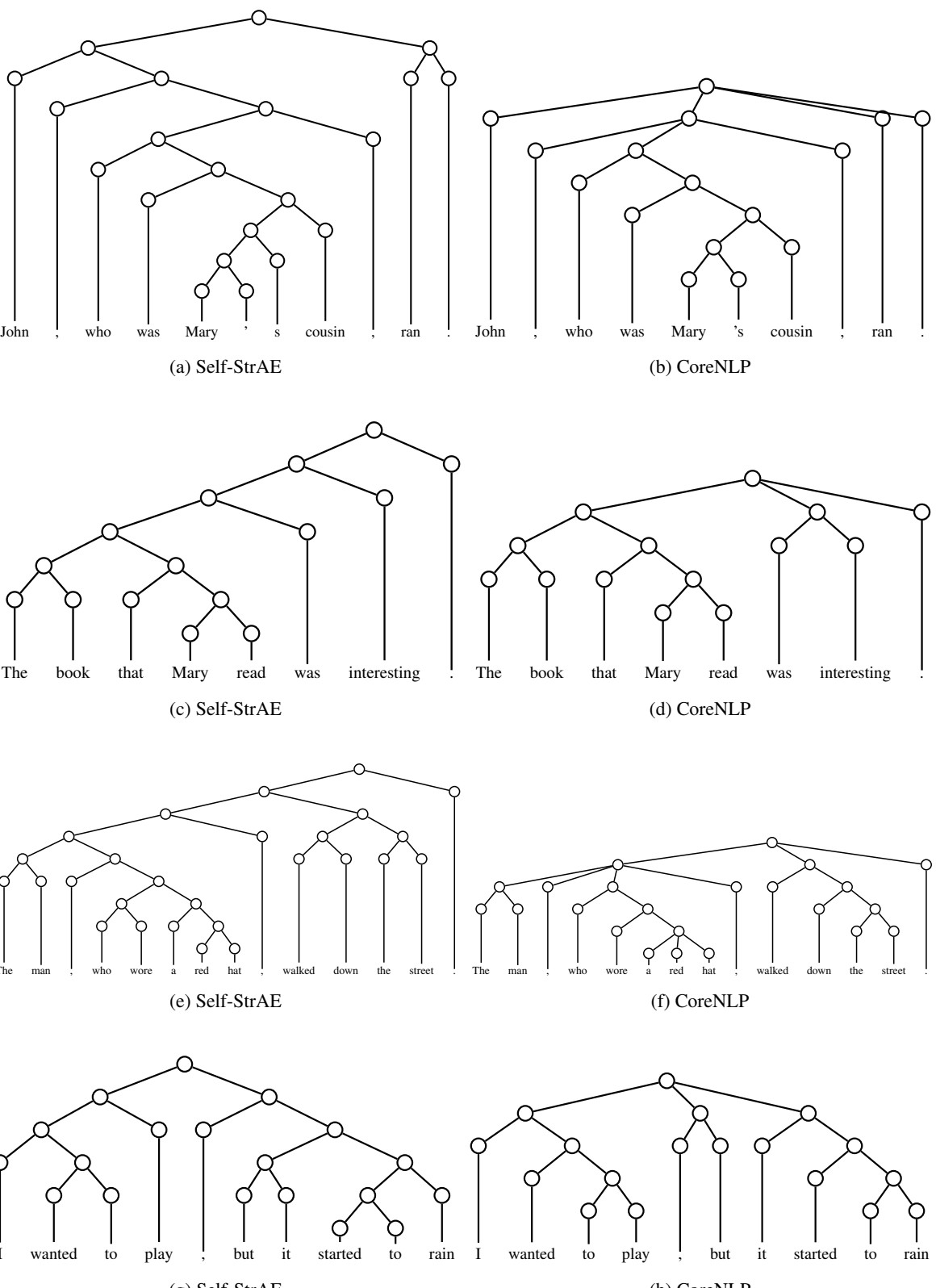

Figure 5: Comparison of trees: Self-StrAE vs. CoreNLP

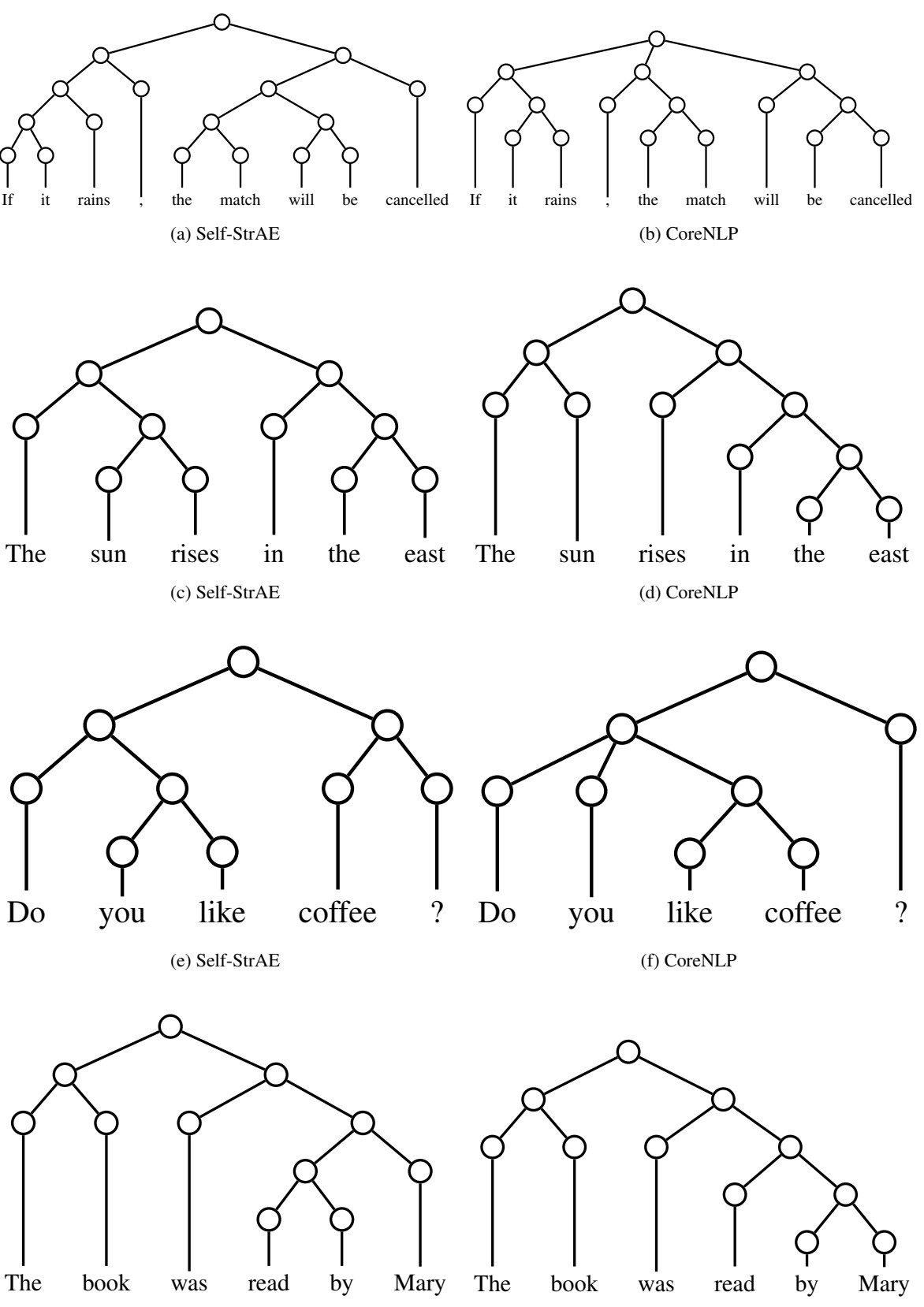

(a) Self-StrAE      (b) CoreNLP

(c) Self-StrAE      (d) CoreNLP

(e) Self-StrAE      (f) CoreNLP

(g) Self-StrAE      (h) CoreNLP

Figure 6: Comparison of trees: Self-StrAE vs. CoreNLP

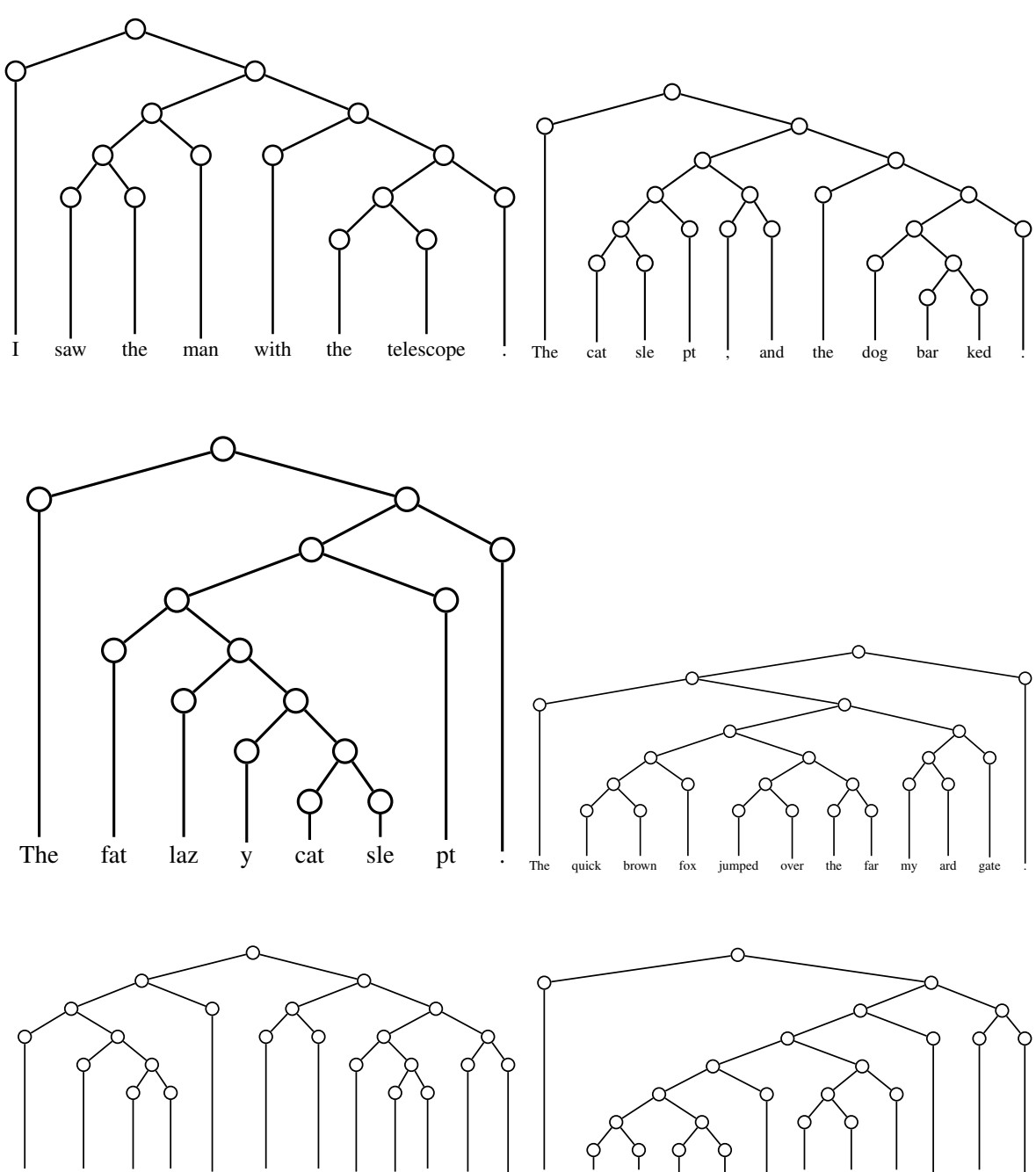

Figure 7: Incomprehensible Cases

Table 7: Comparison Across Structure Type

| Model | Simlex | Wordsim S | Wordsim R | STS 12 | STS 16 | STS B | SICK R | MRPC | RTE |
|---|---|---|---|---|---|---|---|---|---|
| StrAE Syntactic C | 15.54 ± 0.25 | 56.1 ± 0.47 | 43.77 ± 1.01 | 34.59 ± 2.58 | 50.4 ± 0.97 | 41.83 ± 2.23 | 50.3 ± 0.88 | 67.24 ± 0.39 | 56.16 ± 1.59 |
| StrAE Syntactic CE | 17.57 ± 1.08 | 50.72 ± 2.97 | 36.73 ± 6.19 | 5.02 ± 1.1 | 25.86 ± 1.03 | 7.76 ± 0.95 | 39.06 ± 1.51 | 67.48 ± 0.3 | 53.6 ± 0.46 |
| IORNN Syntactic C | 1.9 ± 0.25 | 27.29 ± 0.57 | 11.29 ± 0.5 | -4.2 ± 0.97 | 17.47 ± 1.14 | 1.87 ± 0.55 | 30.48 ± 0.55 | 67.08 ± 0.16 | 55 ± 1.44 |
| IORNN Syntactic CE | 24.36 ± 0.6 | 57.53 ± 1.59 | 36.55 ± 1.78 | -0.17 ± 0.92 | 22.68 ± 0.98 | 5.04 ± 0.55 | 38.41 ± 0.63 | 67.24 ± 0.63 | 54.28 ± 0.55 |
| LSTM Syntactic C | 6.45 ± 0.4 | 37.39 ± 1.2 | 20.6 ± 1.49 | -1.34 ± 1.33 | 23.95 ± 0.45 | 4.25 ± 1.11 | 32.9 ± 0.55 | 68.16 ± 0.32 | 52.76 ± 1.09 |
| LSTM Syntactic CE | 14.23 ± 2.01 | 48.7 ± 3.22 | 33.24 ± 3.02 | -1.66 ± 1.29 | 19.38 ± 1.21 | 6.7 ± 1.04 | 34.55 ± 1.15 | 68 ± 0.0 | 52.2 ± 1.98 |
| StrAE RB C | 4.6 ± 2.5 | 19.6 ± 9.0 | 13.4 ± 9.2 | 3.7 ± 4.0 | 17.9 ± 10.9 | 3.11 ± 5.36 | 22.65 ± 2.88 | 66.32 ± 0.44 | 50.2 ± 1.77 |
| StrAE RB CE | 7.76 ± 2.23 | 9.79 ± 4.3 | 2.85 ± 7.76 | 9.94 ± 1.86 | 14.17 ± 1.95 | 3.87 ± 2.47 | 26.77 ± 1.41 | 62.24 ± 9.71 | 51.4 ± 1.48 |
| IORNN RB C | 3.66 ± 0.21 | 28.24 ± 0.75 | 8.7 ± 3.73 | 2.92 ± 1.31 | 18.5 ± 0.95 | -2.47 ± 1.23 | 24.6 ± 1.67 | 67.15 ± 0.26 | 52.2 ± 0.88 |
| IORNN RB CE | 23.27 ± 0.416 | 61.79 ±1.72 | 40.42 ± 3.02 | 8.85 ±0.75 | 24.79 ± 0.2 | 3.38 ±0.49 | 25.53 ± 0.79 | 67 ± 0.17 | 53 ± 0.54 |
| LSTM RB C | 9.58 ± 0.76 | 40.97 ± 0.94 | 22.87 ± 0.4 | 6.96 ± 0.77 | 32.17 ± 0.43 | 2.76 ± 0.31 | 29.65 ± 0.31 | 68.36 ± 0.39 | 53.6 ± 0.75 |
| LSTM RB CE | 8.69 ± 2.21 | 31.62 ± 3.79 | 19.43 ± 5.84 | 12.67 ± 2.85 | 18.65 ± 3.06 | 5.87 ± 3.51 | 28.67 ± 0.53 | 68.16 ± 0.32 | 53.12 ± 1.31 |
| StrAE BB C | 1.0 ± 4.9 | 43.4 ± 16.9 | 33.6 ± 12.6 | 16.6 ± 3.0 | 16 ± 4.6 | 9 ± 3.0 | 24.18 ± 0.32 | 68 ± 0.0 | 53.5 ± 0.0 |
| StrAE BB CE | 13.59 ± 2.15 | 52.4 ± 1.7 | 35.08 ± 2.42 | 3.46 ± 1.86 | 19.66 ± 1.34 | -0.37 ± 1.75 | 24.99 ± 0.43 | 61.28 ± 0.57 | 52.56 ± 0.84 |
| IORNN BB C | 2.2 ± 0.85 | 8.39 ±2.85 | 2.64± 1.86 | 0.77 ± 0.55 | 10.9 ± 0.09 | 7.67 ± 0.57 | 20.88 ± 0.37 | 67.48 ± 0.35 | 53.24 ± 0.43 |
| IORNN BB CE | 19.02 ± 0.48 | 57.14 ± 0.91 | 44.92 ± 0.98 | 9.46 ± 0.58 | 18.11± 0.81 | 8.86 ±0.91 | 24.3 ± 0.93 | 67.4± 0.36 | 55.36 ± 1.06 |
| LSTM BB C | 4.28 ± 0.65 | 28.17 ± 1.32 | 18.72 ± 1.91 | 10.51 ± 0.54 | 10.75 ± 0.44 | 1.05 ± 0.18 | 23.32 ± 0.33 | 68.0 ± 0.00 | 53.00± 0.70 |
| LSTM BB CE | 4.67 ± 2.78 | 21.21 ± 6.27 | 14.04 ± 4.78 | 8.13 ± 2.18 | 12.99 ± 1.77 | 9.90 ± 9.90 | 21.15± 1.43 | 68.0 ± 0.00 | 53.20 ± 0.81 |