# OpenReview forum: "StrAE: Autoencoding for Pre-Trained Embeddings using Explicit Structure"
_EMNLP/2023/Conference — EMNLP 2023 Main_

### Official Review · Reviewer_xyAG · 2023-08-04

**Soundness:** 4

**Excitement:**

4: Strong: This paper deepens the understanding of some phenomenon or lowers the barriers to an existing research direction.

**Missing References:**

The way Self-StrAE build the tree bottom-up following the similarity is reminiscent of the Easy-First strategy, which can be added to a section for related work.


**Paper Topic And Main Contributions:**

This paper presents an auto-encoder architecture for sentences and words where the hierarchical structure is considered an explicit bias.
The model computes two representations for every node of the binary parse tree corresponding to the input sentence: one when traversing the tree bottom-up from leaves to root which is basically a compression of the information contained in the word embeddings following preselected paths, and the other one is computed when traversing back the tree in a top-down fashion where the information compressed at the root level is expanded recursively in the children nodes down to the leaves where tokens are predicted back.

The model is based on the notions of faithfulness, ie. the representation of a node has only access to its children nodes (resp. parent node) during the bottom-up (resp. top-down) phase. This is in contrast with previous propositions such as IORNNs or Tree-LSTMs.
This seems to be a crucial point as is demonstrated by the new training method based on constrastive-learning, where each bottom-up node representation
must be more similar to its top-down counterpart than any other node representation.
StrAE can take really advantage of this form of learning while IORNNs and Tree-LSTMs have more mitigated results.

The comparisons with IORNNs and Tree-LSTMs are performed on various tasks and show that StrAE outperforms the baselines, and moreover that the advantage is stronger when the parse trees are linguistically motivated foir sentence-level semantics tasks.
Then the comparisons with word-only encoders, such as FastText and RoBERTa, show that the hierarchical information gives a strong inductive bias: with very few parameters the system is able to outperform the RoBERTa baseline, both in small and large data regimes.


The paper is really well written: the model is well presented and advocated for, the experimental results look solid.


**Questions For The Authors:**

- Q1 training and decoding times are  not given, nor compared with other methods. Can you discuss them? I think it would be a nice addition to the paper.
- Q2 in Section 4 have you tried  experiments regarding the different tree shapes with really inconsistent trees ? (ie. at the beginning of the training or at each epoch, draw a random tree structure for each sentence) to see how the models behave when the merge are really inconsistent. I disagree over the fact that right-branching or balanced structures are inconsistent, merely linguistically inconsistent.
  Moreover, the tree shapes produced by Self-StrAE are not discussed enough: how do they look? linguistically plausible, right-branching, balanced... or random?
- Q3 using the NSP objective during transformer pretraining could enable the use of the CLS token as sentence representative instead of the simplistic average, and could lead to better performance. Have you tried? In other words, maybe the transformer baseline for sentence-level tasks is artificially low.


**Reasons To Accept:**

- Again, the paper is really well written: the model is well presented and advocated for, the experimental results look solid.
- Moreover, the limited number of parameters used in this model, could be of interest outside the "interpretability" community, as an example of a resource-friendly architecture (but see question Q1 below)


**Reasons To Reject:**

I can't see one after my reading (unless training and decoding times are prohibitively slow, see Q1)


**Reproducibility:**

4: Could mostly reproduce the results, but there may be some variation because of sample variance or minor variations in their interpretation of the protocol or method.

**Reviewer Confidence:**

4: Quite sure. I tried to check the important points carefully. It's unlikely, though conceivable, that I missed something that should affect my ratings.

---

> ### Author Rebuttal · Authors · 2023-08-29
>
> Thank you for your positive feedback, insightful suggestions and comments.
>
> **A1 Time**:
> The training and inference speeds are not prohibitively slow. We report below iterations per second on a single A40 GPU over a batch size of 128.
> Training an epoch takes ~10m for the StrAE variants and ~4m for RoBERTa, although RoBERTA needs to train for significantly longer to maximise performance—needing 100 epochs vs 15 for the StrAE variants. Total training time on 10M tokens data amounted to ~2.5hrs for the StrAE variants and ~7hrs for RoBERTa.
>
> Note that the speed for StrAE variants is dependent on the structure, with the tree depth dictating the number of sequential steps. The results below for StrAE are for the constituency parses, with balanced binary and right-branching being faster and slower respectively.
> Self-StrAE during encoding has a disadvantage in this regard as it only merges once per step.
> In practice, this could also be improved by merging the top-k most similar nodes, rather than the argmax at each step.
>
>
> | Model     | Train Speed GPU | Eval Speed GPU | Train Speed CPU | Eval Speed CPU |
> |-----------|-----------------|----------------|-----------------|----------------|
> | StrAE     | 8.55 it/s       | 9.09 it/s      | 2.20 s/it       | **1.39 it/s**      |
> | IORNN     | 7.67 it/s       | 8.69 it/s      | 2.26 s/it       | 1.24 it/s      |
> | Tree-LSTM | 6.38 it/s       | 7.43 it/s      | 2.44 s/it       | 1.23 it/s      |
> | Self-StrAE| 7.59 it/s       | 9.46 it/s      | **1.85 s/it**       | 1.37 it/s      |
> | RoBERTa   | **18.2 it/s**       | **43.54 it/s**     | 11.12 s/it      | 3.37 s/it      |
>
>
> **A2a Truly random trees**:
> Your comment about the kind of inconsistency right-branching and balanced trees provide vs truly random trees is indeed correct.
> We are currently running experiments to test the performance of random trees, which are slightly complicated by two factors: (i) ensuring that the trees are indeed random, i.e., not inadvertently picking up on low-level patterns, and (ii) time and compute to run the full model as was done with the other tree types. We hope to be able to update this response with results as soon as we have them, and of course will include them in the updated manuscript.
>
>
> **A2b Self-StrAE Trees**: We also wholeheartedly agree these should be discussed in the paper. We will update the manuscript with some qualitative examples and descriptive statistics. It doesn’t seem possible for us to share the qualitative examples in the rebuttal, but we can share the statistics.
>
> These statistics come from the average across the development split of our 10M token pre-training set, which consists of 40k sequences with an average length of 23.58. Self-StrAE trees have an average depth of 9 levels, compared with 23 (if we round up for simplicity) for right-branching trees, and 5 for balanced binary trees. Self-StrAE trees exhibit a slight preference towards right-branching factor, with each non-leaf node on average having fewer left than right successors 60% of the time. We calculate branching preference by looking at the depth of the subtree for the left and right child of each node.
>
> Qualitatively, the trees are certainly not random, nor purely right-branching, and also, perhaps unsurprisingly, not aligned with known types of linguistic trees.
> A qualitative analysis on a small set of example sentences reveals the emergence of some sensible patterns. For example, the model learns to merge determiner-noun, or determine-adjective-noun tuples into a single constituent. It also appears to have learned to segment sentences that contain conjunctions (e.g. The fat cat slept _and_ the angry dog barked), into the two separate sub phrases.
>
> However, there are definite limitations to the linguistic plausibility of the trees. We noticed that when the number of intervening adjectives between a determiner and a noun increases, the model struggles to resolve the determiner. For example, in the phrase ‘The fat lazy tired old cat’, the model seeks to merge 'cat' and all the corresponding adjectives, but then fails to merge the determiner and complete the constituent.
>
> Given the simplicity of using the greedy agglomerative (easy-first) clustering based on cosine similarity to determine merge order, this behaviour is not entirely unexpected. The structure-induction process would need to be made more flexible to better handle long-distance dependencies. As stated in the limitations section, we didn’t expect or really intend for Self-StrAE to learn something resembling tree bank syntax, just the best compression order w.r.t its data and training objective and subject to its limitations. It would be interesting for future work to compare how these patterns change with a more sophisticated technique for determining merge order, or modifications to the training objective.
>
> We thank the reviewer for the suggestion, and would welcome any feedback on particular behaviours we should probe for that they think might be of interest.
>
>
> **A3 NSP**: The reason we use mean pooling and MLM is due to precedents and findings established in prior work. S-BERT [1] demonstrates that mean pooling is significantly more effective than using CLS for unsupervised STS, and these findings are easy to replicate using pretrained models from Huggingface. It is also the strategy the authors used for their model, which achieved SOTA sentence embeddings. In the RoBERTa paper [2] the authors demonstrate that completely removing NSP as a pre-training objective ‘matches or slightly improves downstream task performance’. Secondly, MLM only means that the RoBERTa baseline has greater parity to the objectives that Self-StrAE and Fasttext are trained with. As neither of these models use additional labels during pretraining. Given these findings, we believe our decisions for the RoBERTa baselines are fair and justified.
>
> Thank you for pointing out the missing reference; we agree our method is similar, and will add a discussion to it in the related work section of our updated manuscript. The two papers we believe are most appropriate are [3,4] and we will include them in the updated manuscript.
>
>
> [1] [Sentence-BERT: Sentence Embeddings using Siamese BERT-Networks](https://aclanthology.org/D19-1410) (Reimers & Gurevych, EMNLP-IJCNLP 2019)
>
> [2] [RoBERTa: A Robustly Optimized BERT Pretraining Approach](https://arxiv.org/abs/1907.11692) (Liu et al. 2019)
>
> [3] [An Efficient Algorithm for Easy-First Non-Directional Dependency Parsing](https://aclanthology.org/N10-1115) (Goldberg & Elhadad, NAACL 2010)
>
> [4] [Partial parsing via finite-state cascades](https://www.cambridge.org/core/journals/natural-language-engineering/article/abs/partial-parsing-via-finitestate-cascades/431C1780A3D3BED2E42D79B1D14C6861) (Abney S. 1996)

---

### Official Review · Reviewer_D1Qk · 2023-08-04

**Soundness:** 3

**Excitement:**

3: Ambivalent: It has merits (e.g., it reports state-of-the-art results, the idea is nice), but there are key weaknesses (e.g., it describes incremental work), and it can significantly benefit from another round of revision. However, I won't object to accepting it if my co-reviewers champion it.

**Paper Topic And Main Contributions:**

This paper proposes a Structured AutoEncoder framework (StrAE) that uses hierarchical
compositions explicitly to produce multi-level representations, and it adopts an encoder-decoder architecture with contrastive loss. Moreover, to avoid using structure information as input, they proposed a variant termed Self-StrAE that is able to learn how to compose each token.
Experimental results show that the proposed framework outperforms baselines that don’t involve explicit hierarchical compositions, and is comparable to models given informative structure.

**Questions For The Authors:**

1. There is no definition for the notation N when it first appears
2. In Table 4, could you give the total number of parameters for all the competitors
3. Sentence embeddings produced by the average output of BERT-like models perform much worse than GolVe (46.35 vs 46.35 on the STS benchmark [1]). Therefore the comparison is not fair enough.
4. RoBERTa is not designed for context-free word embeddings. Are there any empirical studies to show the way you induce word-level embeddings from language models is feasible?
5. Compared to a tiny version of BERT (2 layers, 2 heads, 128 dimensions), the performance of RoBERTa in Table 4 is much lower (MRPC: 81.1 vs 69.19; RTE: 57.2: 54.86) [2].
6. Can you provide another naive baseline in Table 4, i.e., CNN (or) with Fasttext or GloVe? I'd like to see the benefit of introducing explicit compositions
7. In contrastive loss, the objective is to maximize the similarity between $\overline{e_i}$ and $\underline{e_i}$ while minimizing the similarity between other token embeddings (in-batch negatives). Does this way introduce noises? For example, two identical words (or semantically related words) appear in the same sentence.
8. In Table 3, could you describe the model size of each competitor?

[1] Sentence-BERT: Sentence Embeddings using Siamese BERT-Networks

[2] https://github.com/google-research/bert

**Reasons To Accept:**

1. The motivation is clearly stated and the paper is easy to follow
2. Using auto-encoder and contrastive loss to inject structural information is interesting
3. StrAE can outperform other tree models (IORNN and Tree-LSTM), which shows the effectiveness of the proposed method

**Reasons To Reject:**

My main concern is the experiment design of section 5 (Comparison to Unstructured Baselines):
1. This work averages the RoBERTa final output layer to derive sentence-level embeddings. However, this way yields very bad embeddings that are even worse than averaging GloVe embeddings, as discussed in the paper of Sentence-BERT.
2. The power of RoBERTa is to produce contextual embeddings, and the work uses static embeddings induced from RoBERTa for comparison, which might result in misleading conclusions

My another concern is the main benefit of using StrAE but I did not see it in the current version
1. Does the framework consume fewer parameters or have a faster inference speed
2. Can we scale up the model to a very large size for having powerful reasoning capability and general-purpose usage
3. Can we integrate StrAE into pre-trained language models as an auxiliary task


**Reproducibility:**

4: Could mostly reproduce the results, but there may be some variation because of sample variance or minor variations in their interpretation of the protocol or method.

**Reviewer Confidence:**

4: Quite sure. I tried to check the important points carefully. It's unlikely, though conceivable, that I missed something that should affect my ratings.

**Typos Grammar Style And Presentation Improvements:**

I feel a bit confused by the notations in the paper and cannot distinguish a scalar, vector, and matrix easily.

---

> ### Author Rebuttal · Authors · 2023-08-29
>
> Thank you for your detailed feedback, suggestions, insightful questions, and positive appraisal of our work. We would like to begin our rebuttal by addressing the concerns raised by the reviewer before moving to the questions.
>
> Regarding the use of RoBERTa and unstructured baselines, we note that
> 1. **RoBERTa for sentence-level embeddings**:
>     Averaging the final layer (i.e. mean pooling)  is demonstrably the best approach for
>          unsupervised STS with BERT like models, as shown and used by S-BERT [1].
>     At lower dataset sizes (as with our work) averaged GloVe embeddings underperform
>          StrAE and RoBERTa (please see evidence in A3 below).
> 2. **Roberta for static embeddings:**
>     We use the same method for extracting static lexical embeddings as established by
>           Vulic et al. 2020 [2].
>      We further provide Fasttext embeddings as a baseline (Table 4 of the paper) which is
>          designed specifically to produce static embeddings.
>
> Regarding the benefit of using StrAE, yes, StrAE does indeed need (much) fewer non-embedding parameters (please see response to A2 below).
>
> As for the question of scalability to very-large sizes, in principle, especially with Self-StrAE, we do not see any specific reason why our method should _not_ be applicable at scale—the only practical constraints, as with most similar methods, is GPU memory.
> As for effectiveness at reasoning or general-purpose use, we suspect that being able to do so could be a fruitful avenue for future exploration, and are excited by the potential of developing a more interpretable and low resource method for this purpose.
>
> Regarding the use of StrAE as an auxiliary task for pretrained models, we agree that it is an interesting question. However, our goal in this work is to explore the use and utility of explicit structure, and combining StrAE with pretrained models would make it harder for us to assign credit properly. We do believe it could prove to be an interesting direction for future work, though.
>
> Typos/Grammar/Presentation:
> We will clarify notation by employing normal-weight lower-case letters as scalars, bolded lower-case letters as vectors and upper-case letters as matrices.
>
> We now respond to the specific questions raised in the review:
>
> **A1: Notation N**
> Apologies for the oversight. The use of N (L147) refers to the square root of the embedding dimension. Each embedding is NxN as we treat them as matrices during composition and decomposition. In our experiments, each StrAE embedding is a 10x10 matrix.
>
> **A2: Parameter Counts**
> Thank you for the suggestion; we agree that these should be included in the paper and will amend the manuscript accordingly.
> We also need to correct the number of parameters for the RoBERTa baseline, which was reported as 1.5 million (Line 485-487) —the true number is closer to 4 million, and reported in the table below. We will fix the error in the manuscript. The list of parameters for Q2 and Q8 is as follows:
>
> | Model Name      | Parameters excluding embedding matrix (and LM head where applicable) |
> |-----------------|-----------------------------------------------------------------------|
> | StrAE/Self-StrAE| 400                                                                   |
> | IORNN           | 40,400                                                                |
> | Tree-LSTM       | 260,600                                                               |
> | RoBERTa         | 3,950,232                                                             |
>
> We omit Fasttext from the table, as the number of parameters effectively just corresponds to the size of the embedding matrix. We do not include the embedding matrix in counting parameters in the table, as is standard practice, as it only reflects the type of tokenisation used and not the model complexity.
>
> **A3: RoBERTa vs GLoVe Sentence Embeddings**
> You are indeed correct that the S-BERT paper notes that mean-pooled embeddings of BERT-like models underperforms GloVe. However, this does not appear to be the case with smaller-scale pre-training data. To demonstrate this, we evaluate Self-Strae, RoBERTa, and GloVe on the STS eval tasks with our 10M and Wiki103 datasets.
>
> | Model                  | STS-12       | STS-16       | STS-B        | SICK-R       |
> |------------------------|--------------|--------------|--------------|--------------|
> | Self-StrAE BPE 10M     | 34.42 +- 0.73| 49.93 +- 0.38| 36.68 +- 0.64| 51 +- 0.3    |
> | RoBERTa 10M            | 29.48 +- 3.28| 50.88 +- 1.11| 38.36 +- 1.9 | 49.58 +- 0.91|
> | GloVe 10M              | 25.08 +- 0.95| 27.11 +- 0.4 | 21.84 +- 0.3 | 46.09 +- 1.22|
> | Self-StrAE BPE Wiki103 | 46.64 +- 0.23| 52.08 +- 0.37| 40.59 +- 0.83| 51.94 +- 0.4 |
> | RoBERTa Wiki103        | 35.38 +- 1.47| 52.64 +- 1.11| 39.74 +- 39.74| 50.68 +- 0.43|
> | GloVe Wiki103          | 36.8 +- 1.28 | 31.11 +- 0.49| 29.83 +- 0.71| 51.35 +- 0.87|
>
> In both these settings, averaged GloVe vectors underperform those from Self-StrAE and RoBERTa, although the gap with RoBERTa diminishes on Wiki103. As a further sanity check, we have also compared the full 840B GloVe embeddings with RoBERTa base, and can replicate the findings of the S-BERT paper.
>
> In the context of the data scales investigated in our paper, RoBERTa proves a stronger baseline. We are however happy to include GloVe as a baseline in the updated manuscript, and thank the reviewer for raising the issue, as it is possible other readers might share similar concerns.
>
>
> **A4: RoBERTa and static word embeddings**
> Yes indeed; we employ the technique derived in [2] to construct static embeddings from RoBERTa’s contextual embeddings, noting that the chosen method is extensively empirically validated therein.
>
> **A5: Comparison with Tiny BERT**
> There are two notable differences between our RoBERTa and Tiny-BERT.
> 1. Tiny-BERT is fine-tuned on the downstream task while all our models are evaluated frozen.
> 2. Tiny BERT is pre-trained on significantly more data: Wikipedia + Books contains 3.3 billion tokens, whereas our two pre-training sets consist of 10 and 103 million tokens respectively.
>
> Regarding MRPC, Table 5 in S-BERT [1] gives the MRPC score for averaged BERT embeddings as 69.45 which corresponds to our results.
>
> **A6: Additional Naive Baseline**
> Note that most of our evaluation is zero-shot—simply computing measures with learned embeddings. To introduce a CNN baseline would require pre-training from scratch (unless we have misunderstood what you were requesting?).
> To our knowledge, there isn’t an established model/architecture that employs CNNs directly for pre-training on natural language, and the plethora of hyperparameter choices involved in designing something reasonable here would not necessarily tell us if performance was because of/despite the choices made. If there is a specific model you had in mind, please do let us know, and we’d be happy to add it as a baseline.
>
> However, to address your more general point about a baseline that does not _explicitly_ model hierarchical composition, we can attempt to answer that by employing a standard Bi-LSTM framework on our data (10M tokens). We provide evaluation results for this below, and will be happy to include this as an additional baseline to Table 4.
>
> |  Model     | Bi-LSTM BPE   | Bi-LSTM WORD  |
> |---------------|---------------|---------------|
> | Simlex        | 12.88 +- 0.77 | 23.74 +- 1.17 |
> | Wordsim S     | 39.46 +- 2.17 | 61.94 +- 2.44 |
> | Wordsim R     | 32.22 +- 3.07 | 41.46 +- 1.73 |
> | STS-12        | 9.22 +- 0.87  | 11.18 +- 1.1  |
> | STS-16        | 33.78 +- 1.66 | 34.86 +- 0.86 |
> | STS-B         | 14.06 +- 2.01 | 13.46 +- 1.11 |
> | SICK-R        | 40.36 +- 0.27 | 42.36 +- 0.19 |
> | MRPC          | 68.21 +- 0.41 | 68.95 +- 0.38 |
> | RTE           | 53.90 +- 0.96 | 54.09 +- 0.95 |
> | Score         | 33.79 +- 0.64 | 39.12 +- 0.34 |
>
> **A7: Noise in Contrastive Loss**
> Yes, you are indeed correct to note that the contrastive loss can include some measure of noise through false negatives (FN, i.e. same embedding being counted as negative by appearing elsewhere in the batch).
>
> The noise, in practice, appears to have minimal effect, and performance is largely unaffected without taking mitigation measures. This is primarily due to the effect of many more _true_ negatives (TN) than false negatives (FN). However, without taking any measures, learning plateaus quicker and results are marginally lower. The issue can be mitigated by masking false negatives, so they don’t affect the softmax and don’t count toward loss. We will add an appendix to the updated manuscript with ablations showing the effect of masking to mitigate this issue.
>
> [1] [Sentence-BERT: Sentence Embeddings using Siamese BERT-Networks](https://aclanthology.org/D19-1410) (Reimers & Gurevych, EMNLP-IJCNLP 2019)
>
> [2] [Probing Pretrained Language Models for Lexical Semantics](https://aclanthology.org/2020.emnlp-main.586) (Vulić et al., EMNLP 2020)

---

### Official Review · Reviewer_pRFX · 2023-08-05

**Soundness:** 4

**Excitement:**

4: Strong: This paper deepens the understanding of some phenomenon or lowers the barriers to an existing research direction.

**Paper Topic And Main Contributions:**

The paper presents a method for encoding structural information from sentences using an Autoencoder framework. The objective of the approach is to learn token and sentence representations that reflect the hierarchical composition process of language syntax, thus obtaining more expressive token/sentence embeddings. This is achieved through a combination of a recursive Autoencoder architecture and a structure oriented contrastive loss. Two encoding schemes are proposed: one based on external tree structure supervision (StrAE) and one based on self-clustering (Self-StrAE) of embeddings, which are the main contributions of this work. An additional contribution is the empirical analysis of the contrastive objective over different architectural constraints and different tasks.

**Reasons To Accept:**

This work addresses the incorporation of linguistic structural patterns (e.g., constituency) into language representations, by means of an autoencoder model of much smaller scale than current large language models. This is of great interest for the NLP community, not only due to structural information learning being a longstanding goal in NLP and its practical applications, but also due to the lower computational cost of the proposed approach. The text is very well written and straightforward to follow, the experiments have appropriate coverage, and the findings are consistent with the claims and argumentation.

**Reasons To Reject:**

Not a reason to reject, but the embedding process for Self-StrAE needs to be better clarified. Is it the same as StrAE?

**Reproducibility:**

4: Could mostly reproduce the results, but there may be some variation because of sample variance or minor variations in their interpretation of the protocol or method.

**Reviewer Confidence:**

4: Quite sure. I tried to check the important points carefully. It's unlikely, though conceivable, that I missed something that should affect my ratings.

**Typos Grammar Style And Presentation Improvements:**

L.554-555: "explicitly incorporating hierarchical compositions be highly beneficial" -> "explicitly incorporating hierarhical compositions *can* be highly beneficial"

L.631-632: "when we allow for a significantly more flexible models" -> "when we allow for significantly more flexible models"

---

> ### Author Rebuttal · Authors · 2023-08-29
>
> Thank you for the positive feedback and time taken reading the paper.
>
> The embedding process for Self-StrAE is indeed identical to StrAE. The only difference lies in how the order of compositions (i.e. the tree structure) is defined. Self-StrAE orders compositions according to cosine similarity between adjacent node embeddings in the frontier, merging the argmax at each step. Figure 4 of the paper is intended to show this process with the example sentence ‘Homer ate doughnuts’. Following the example in the figure, the similarity between ‘ate’ and ‘doughnuts’ is greater than that of ‘ate’ to ‘Homer’, so the model first merges ‘ate doughnuts’ into a single embedding using the composition function. At the next step, the model merges ‘Homer’ and ‘ate doughnuts’ into a single embedding and arrives at the root.
>
> We save the merge history in an adjacency matrix so that by the time the encoder reaches the root node, the adjacency matrix represents the whole tree over the input sequence. We then convert the adjacency matrix into a DGL graph (DGL is a library for graph ML), and pass that as input to the decoder. The decoder operates exactly the same as in vanilla StrAE, only that in this case the input graph is defined by the encoder’s merge history not e.g. a syntactic parser.
>
> We hope this explanation helped make things clearer, and will update the manuscript with a more detailed description.
>
> Thank you also for your pointers to typos and suggestions for improving the text—we will incorporate these in the updated manuscript.

---

### Meta-Review · Area_Chair_GucX · 2023-09-19

**Recommendation:** 4

**Metareview:**

This paper proposes an encoder-decoder network to compositionally encode by explicitly factoring in constituency parses for sentences. Results are positive on a number of textual similarity and inference tasks. The reviews converge in their soundness and excitement scores.

---

### Decision · Program_Chairs · 2023-10-07

**Decision:**

Accept-Main

**Comment:**

This paper proposes an encoder-decoder network to compositionally encode by explicitly factoring in constituency parses for sentences. Results are positive on a number of textual similarity and inference tasks. The reviews converge in their soundness and excitement scores.